# Effects of Hard Water Boiling on Chalky Rice in Terms of Texture Improvement and Ca Fortification

**DOI:** 10.3390/foods12132510

**Published:** 2023-06-28

**Authors:** Sumiko Nakamura, Ken’ichi Ohtsubo

**Affiliations:** Faculty of Applied Life Sciences, Niigata University of Pharmacy and Applied Life Sciences, 265-1, Higashijima, Akiha-ku, Niigata 956-8603, Japan; snaka@nupals.ac.jp

**Keywords:** chalky rice, xylanase activity, cellulase activity, hard water, hardness

## Abstract

In the present paper, we investigated the characteristics of chalky rice grains generated by ripening under high temperature and compared them with whole grains. We evaluated 14 unpolished *Japonica* rice grains harvested in Japan in 2021, and these samples (original grains) were divided into two groups (a whole grain group and a chalky grain one). We found that not only activities of endogenous amylase and proteinase, but also cell wall-degrading enzymes, such as xylanase and cellulase, changed markedly between chalky grains and whole grains. Using rice grains blended with 30% of chalky grains as the material, we compared the sugar and mineral contents and textural properties of the rice grains soaked and boiled in either ordinary water or hard water, such as Evian or Contrex. It was shown that xylanase, in addition to amylase and proteinase, may play an important role in changing the texture of the boiled chalky rice grains. For the sake of preventing the above-mentioned deterioration in the texture of boiled grains of chalky rice, we tried to use hard water, such as Evian or Contrex, to soak and cook the chalky rice grains. It was shown that the hard water was useful for the prevention of texture deterioration of the boiled rice grains due to inhibition of the activities of endogenous hydrolytic enzymes, such as α-amylase, β-amylase, proteinase, and xylanase. Furthermore, we found that the hard water was useful in increasing the calcium absorption through the meal by 2.6 to 16.5 times.

## 1. Introduction

Rice (*Oryza sativa* L.) is one of the three most important cereal crops in the world, along with corn and wheat, and supplies calories to about half of the world’s population. Rice is widely grown in over 100 countries [1]. Global warming is the most serious environmental issue, and high temperature stress during the rice ripening period causes deterioration not only in the grain yield, but also in quality, by generating chalky grains [2,3]. Global warming has serious implications for our future. Mitsui et al. [4], Asaoka et al. [5], and Nakata et al. [6] have reported that high temperatures cause the inhibition of starch synthases and the activation of α-amylase.

The endosperm starch of rice grains is damaged by high-temperature ripening, which leads to the lowering of the food quality by factors such as the hardness and stickiness of boiled rice grains and the pasting properties of rice flours [7,8]. In our previous paper, we reported that there are marked differences in properties between chalky grains and whole grains using 54 *Japonica* unpolished rice grains damaged severely or lightly by high-temperature ripening [9]. We reported that the α-amylase activity, proteinase activity, and n-6/n-3 ratio of polyunsaturated fatty acid [10,11] of whole grains were significantly lower than those of chalky rice grains. Furthermore, the chalky rice grains, after boiling, were different from whole grains not only in their physical properties, but also in taste components such as oligo saccharides and amino acids, which affect the eating quality of boiled rice [9]. We also made it possible to estimate the degree of damage to rice grains ripened under high temperatures using the pasting properties measured by an RVA as explanatory variables [9,12].

The texture of boiled rice grains is markedly affected by the cell wall of endosperm [13]. Boiled rice grains become softer with the degradation of the cell wall, which suggests the important role of cell wall-degrading enzymes. For examples, softening of the texture by the addition of cellulase [14] and xylanase [15] on boiling were reported in [16]. We formerly reported that endogenous xylanase and cellulase play important roles in determining the texture of the cooked rice grains, similarly to the amylose content [17]. Furthermore, it seems necessary to measure the amounts of tasty substances, such as oligo saccharides and free amino acids, in addition to the physical properties, such as the hardness and stickiness of the boiled rice grains [18,19,20,21]. Therefore, it is necessary to compare the texture- and taste-related substances in boiled rice for the purpose of elucidating the difference in palatability between whole grains and chalky grains [10,22,23]. There are few scientific reports about the relationship between high-temperature damage of rice grains and changes in proteinase activity, amino acids, and fatty acid composition [24,25].

Recently, rice industries, such as rice catering services, rice cracker makers, and rice wine brewers, have suffered from the high-temperature ripening of rice grains, as high-temperature-damaged rice grains have textures too soft and too sticky for the rice processing procedure [20,26]. Therefore, it is necessary to improve the texture of the high-temperature-damaged rice grains. 

Furthermore, rice consumers request not only palatable rice, but also “healthy rice”, such as brown rice, pigmented rice, and pre-germinated brown rice in order to prevent the lifestyle-related diseases by supplying dietary fiber, vitamins, minerals, gamma-amino butylic acid (GABA), etc. [27]. The elderly population has increased markedly in north-eastern Asia, and various kinds of health disorder, such as osteoporosis, have become extremely important problems [28].

In this study, we attempted to elucidate how high-temperature ripening affects the deterioration in quality of chalky grains. Although we reported that not only α-amylase, but also protease, activities are activated by high-temperature-damaged chalky rice grains, we searched for another cause of the quality deteriorations of the chalky rice grains. Cell wall-degrading enzymes, such as cellulase and xylanase, were our novel targets as candidates as the causes of damage to the quality of chalky rice grains. Another aim of this investigation was the development of a method to improve the physical properties of boiled chalky rice grains.

To achieve the above-mentioned objectives, we collected and analyzed 14 *Japonica* rice samples (original grains), which were divided to two groups (a whole grain group and a chalky grain one). A novel method was used to maintain good texture by inhibiting glycolytic enzymes and cell wall-degrading enzymes during the boiling process. We measured the contents of sugars and the textural properties of boiled rice grains boiled after soaking in two types of hard water. Using rice grains consisting of either 100% whole grains or rice blended with 30% chalky grains, we measured the contents of sugars and the textural properties of boiled rice after soaking in two types of hard water.

## 2. Materials and Methods

### 2.1. Materials

The unpolished rice samples were purchased in 2021 at a local market, and were subjected to measurement in 2022 (*Japonica* subspecies) (*n* = 16). These original rice samples were divided manually, based on their apparent chalkiness, into two groups (whole grain and chalky grain): The ordinary *Japonica* rice included *Kumasannokagayaki* (Kumamoto prefecture), *Tsukiakari* (Iwate), *Sasashigure* (Miyagi A), *Sasashigure* (Miyagi B), *Hitomebore* (Miyagi), *Yosakoibijin* (Kochi), *Morinokumasan* (Kumamoto), *Yumeshizuku* (Saga), and *Tsubusuke* (Chiba)(*n* = 9). The high-quality premium *Japonica* rice included *Koshihikari* (Niigata A), *Koshihikari* (Niigata B), and *Koshihikari* (Hyogo C) (*n* = 3). The high-amylose *Japonica–Indica hybrid* rice was *Koshinomenjiman* (Niigata), and the *Indica* rice was *Koshinokaori* (Niigata) (*n* = 2). The types of low-amylose *Japonica* rice were *Milky queen* (Kyoto) and *Yumepirika* (Hokkaido) (*n* = 2). Each sample was stored at 10 °C in a rice storage chamber. 

The Evian (hardness: 304 mg/L, pH: 7.2) and Contrex (hardness: 1468 mg/L, pH: 7.2) for cooking were purchased at a local market in Niigata City. We used purified water (hardness: 17 mg/L, pH: 7.0) as a control. 

### 2.2. Measurement of the Moisture Content of Rice Flour

The moisture contents of the polished and unpolished rice flours were measured using an oven-drying method by drying flour samples of about 2 g for 1 h at 135 °C. An aluminum cup (Wt) and a cup containing the flour sample (Ws), were compared before heating (Wsb) and after heating (Wsa). The moisture content was calculated as follows.
Moisture Content (%) = 100 × ((Wt + Wsb) − (Wt + Wsa))/Wsb

### 2.3. Preparation of Two Kinds of Unpolished Rice Flours

Whole or chalky unpolished rice grains of 14 rice samples—*Kumasannokagayaki* (Kumamoto), *Tsukiakari* (Iwate), *Sasashigure* (Miyagi A), *Hitomebore* (Miyagi), *Yosakoibijin* (Kochi), *Morinokumasan* (Kumamoto), *Yumeshizuku* (Saga), *Tsubusuke* (Chiba), *Koshihikari* (Niigata A), *Koshihikari* (Hyogo C), *Koshinomenjiman* (Niigata), *Koshinokaori* (Niigata), *Milky queen* (kyoto), and *Yumepirika* (Hokkaido)—were used as materials for rice flour. These whole or chalky rice grains were pulverized into rice flours using a cyclone mill (SFC-S1; UDY, Corp., Fort Collins, CO, USA).

### 2.4. Soaking of Polished or Unpolished Rice Flours in 2 Types of Hard Water

Two kinds of rice cultivars, *Koshihikari* (Hyogo, C) and *Tsubusuke* (Chiba), were used for the polished rice samples, and *Koshihikari* (Niigata, A) was used for the unpolished rice. The whole or chalky rice grains of polished rice were soaked in Evian, Contrex, or purified water (tap water treated with a water purifier) at 8 °C for 3 h in refrigerator, respectively, and the unpolished rice was soaked similarly for 48 h. These polished or unpolished rice flours were prepared by pulverizing rice grains after lyophilization (freeze dryer: FD-1, Eyela, Tokyo, Japan). Pulverization was carried out using a cyclone mill (SFC-S1; UDY, Corp., Fort Collins, CO, USA).

### 2.5. α-Amylase Activity

The *α*-amylase activity of the whole or chalky grains of unpolished rice flour (*n* = 14), as well as those of polished rice flours (*n* = 2) and unpolished rice flour (*n* = 1) soaked in Evian, Contrex, or purified water were determined using an enzyme assay kit (Megazyme International Ireland, Ltd., Wicklow, Ireland), as described in our former report [12].

### 2.6. β-Amylase Activity

The *β*-Amylase activity of the whole or chalky grains of unpolished rice flour (*n* = 14) and those of polished rice flours (*n* = 2) and unpolished rice flour (*n* = 1) soaked in Evian, Contrex, or purified water were determined using an enzyme assay kit (Megazyme International Ireland, Ltd.). For measurement of the *β*-amylase activity, rice flour (0.1 g) was extracted with 0.5 mL of extraction buffer, pH 8.0, at 20 °C for 60 min, and was thereafter centrifuged for 10 min at 2000× *g*. The extraction solution (0.6 mL; 6-fold dilution) and substrate (0.1 mL) were preincubated at 40 °C for 5 min. Thereafter, each sample solution was incubated at 40 °C for exactly 10 min, followed by the addition of a stopping reagent (3.0 mL). The absorbance was measured at 400 nm.

### 2.7. Protease Activity

The protease activity of the whole or chalky grains of the unpolished rice flour (*n* = 14), as well as that of the polished rice flours (*n* = 2) and unpolished rice flour (*n* = 1) soaked in two types of hard water or purified water were measured using casein as substrate. Protease activity was measured using the same method described in our former report [12].

### 2.8. Xylanase Activity

The xylanase activity of the whole or chalky grains of unpolished rice flour (*n* = 14), as well as that of polished rice flour (*n* = 2) and unpolished rice flour (*n* = 1) soaked in two types of hard water and purified water were determined using a kit (Megazyme International Ireland Ltd., Wicklow, Ireland). For measurement of the xylanase activity, the rice flour (0.1 g) was suspended in 0.1 M C_2_H_3_NaO_2_ (pH 4.5) buffer containing BSA (0.5 mg/mL) and NaN_3_ (0.02% (*w*/*v*)) at room temperature for 15 min, then centrifuged for 10 min at 1000× *g*. Afterward, the supernatants of the extraction solution (0.05 mL; 2-fold dilution) and substrate solution (0.05 mL) were preincubated at 40 °C for 3 min. Thereafter, each sample solution was incubated at 40 °C for exactly 10 min, followed by the addition of a stopping reagent (Tris-HCl buffer 2% (*w*/*v*)) (pH 10.0) (1.5 mL). The absorbance was measured at 400 nm.

### 2.9. Cellulase Activity

The cellulase activity of the whole or chalky grains of unpolished rice flour (*n* = 14) and that of polished rice flour (*n* = 2) and unpolished rice flour (*n* = 1) soaked in two types of hard water and purified water were determined by a kit (Megazyme International Ireland, Ltd.). For measurement of the cellulase activity, the rice flour (0.1 g) was suspended in 0.1 M C_2_H_3_NaO_2_ (pH 4.5) buffer containing BSA (0.5 mg/mL) and NaN_3_ (0.02% (*w*/*v*)) at room temperature for 15 min, then centrifuged for 10 min at 1000× *g*. After that, the supernatants of the extraction solution (0.2 mL; 2-fold dilution) and substrate (0.05 mL) were preincubated at 40 °C for 3 min. Thereafter, each sample solution was incubated at 40 °C for exactly 10 min, followed by the addition of a stopping reagent (Tris-HCl buffer 2% (*w*/*v*)) (pH 10.0) (3.0 mL). The absorbance was measured at 400 nm.

### 2.10. Polishing and Boiling of Rice Samples

We prepared polished rice (milling yield of 90–91%) using an experimental friction-type rice milling machine (Yamamotoseisakusyo Co., Yamagata, Japan). Samples of 10 g of the blended polished rice grains (blending ratio: whole rice grains:chalky rice grains = 7:3) were added to 16 g (1.6 times, *w*/*w*) of purified water in an aluminum cup as control samples, and another set of rice grains samples (10 g) was added to 16 g (1.6 times, *w*/*w*) of Evian or Contrex [13]. After soaking for 1 h, the samples were boiled in an electric rice cooker (SR-SW 182 National, Japan). The cooked rice samples were kept in the vessel for 2 h at 25 °C, then used for the measurements [13]. The rice samples which we utilized were *Kumasannokagayaki* (Kumamoto), *Tsukiakari* (Iwate), *Sasashigure* (Miyagi A), *Sasashigure* (Miyagi B), *Hitomebore* (Miyagi), *Yosakoibijin* (Kochi), *Morinokumasan* (Kumamoto), *Yumeshizuku* (Saga), *Koshihikari* (Niigata A), and *Koshihikari* (Niigata B) (*n* = 10).

In contrast, the polished rice grains of 100% whole or chalky rice were boiled similarly [13]. The rice samples which we utilized were *Koshihikari* (Hyogo C) and *Tsubusuke* (Chiba).

### 2.11. Measurements of Textural Properties of Boiled Rice Grains

The physical properties of boiled rice grains were measured based on bulk measurements (10 g) and single-grain measurements (the high-compression/low-compression method), which included low compression (compression ratio = 23%: twice), intermediate compression (compression ratio = 46%: twice), and high compression (compression ratio = 92%: twice), according to the 2 × 3 bite method for blended rice. A low compression test (compression ratio = 25%) and a high compression test (compression ratio = 90%) were utilized for 100% whole and chalky rice using a Tensipresser (My Boy System, Taketomo Electric Co., Tokyo, Japan), according to the method described by Okadome et al. [29]. The bulk measurements were repeated 5 times, and single-grain measurements were calculated by measuring 20 individual grains. We used the following parameters for the physical properties of the boiled rice grains for bulk measurement: hardness, toughness, adhesion, stickiness, and cohesiveness [30]. The cohesiveness was measured as A2/A1 for the ratio of hardness with low compression, A4/A3 for the ratio of hardness with intermediate compression, and A6/A5 for the ratio of hardness with high compression. of the parameters used for single-grain measurement were H1 for surface hardness, H2 for overall hardness, S1 for surface stickiness, S2 for overall stickiness, L3 for surface adhesion, S1/H1 (balance H1) for the ratio of stickiness to hardness of the surface layer, S2/H2 (balance H2) for the ratio of stickiness to hardness of the overall layer, A3/A1 (balance A1) for the ratio of adhesiveness to hardness of the surface layer, and A6/A4 (balance A2) for the ratio of adhesiveness to hardness of the overall layer.

### 2.12. Measurement of d-Glucose, Maltose, and Saccharose Contents

The cooked rice flour sample was prepared by pulverization after lyophilization. 

d-glucose, maltose, and saccharose (0.1 g) were extracted from each sample by shaking with 1 mL of 60% ethanol at room temperature for 1 h, and then were measured by the Sucrose/d-Glucose/d-Fructose content assay method (F-kit, Roche/ R-Biopharm AG., Darmstadt, Germany).

### 2.13. Measurement of Color Difference of Boiled Rice after Soaking in 2 Types of Hard Water

The color differences of the boiled blended rice (blend rice: whole rice grains: chalky rice grains = 7:3) soaked in Evian, Contrex, or purified water were measured using a color difference meter (Color Meter NW-11, Nippon Denshoku Co., Tokyo, Japan).

### 2.14. Analysis of Calcium and Magnesium Contents

The calcium and magnesium of the whole and chalky *Koshihikari* (Hyogo C) grains of polished or unpolished boiled rice after being soaked in 2 types of hard water or purified water of whole grains of polished rice of were analyzed by an ICP (inductively coupled plasma) emission spectrometry. The absorbance values were measured as 423 nm for calcium and 285 nm for magnesium. Moreover, the tests on boiled whole and chalky polished *Tsubusuke* (Chiba) rice were analyzed similarly. The measurements of the microminerals in the rice samples were carried out at Japan Food Research Laboratories.

### 2.15. Statistical Analyses

We used Excel Statics (ver. 2006; Microsoft Corp., Tokyo, Japan) for the statistical analysis of the significance of regression coefficients using Student’s t-test, one-way analysis of variance, and Tukey’s test. The method of Tukey’s multiple comparison was statistically analyzed using Excel NAG Statistics 2.0 (The Numerical Algorithms Group, Ltd., Tokyo, Japan).

## 3. Results and Discussion

### 3.1. Activities of Various Hydrolytic Enzymes in Whole or Chalky Unpolished Rice Grains

In addition to amylose content, the cell wall of the rice endosperm also affects the texture of boiled rice grains. When the cellulose of the cell wall is hydrolyzed by cellulase (endo-1,4-beta-d-glucanase), boiled rice grains become softer and stickier [31].

As shown in Figure 1, the endo-xylanase activities of chalky unpolished rice grains of premium *Japonica* rice Koshihikari (*n* = 2), ordinary *Japonica* rice (*n* = 6), low-amylose *Japonica* rice (*n* = 2), and high-amylose *Japonica–Indica* hybrid rice (*n* = 1) were significantly higher than those of whole rice grains. Two samples of ordinary *Japonica* rice and *Indica* rice showed similar tendencies. As a result, whole unpolished rice grains were shown to have significantly lower endo-xylanase activities than chalky unpolished rice grains.

The ratio of the endo-xylanase activities of chalky unpolished rice grains to those of whole unpolished rice grains in low-amylose rice (1.38 ± 0.2) was extraordinary higher than those of premium Koshihikari (1.24 ± 0.1) rice and ordinary *Japonica* rice cultivars (1.20 ± 0.1). Chalky unpolished rice grains showed markedly higher endo-xylanase activities, by 1.1–1.5 times, compared to whole unpolished rice grains in 12 *Japonica* rice variants in 2021.

It was found that the endo-xylanase activity of whole unpolished rice grains is lower than that of chalky unpolished rice grains. We also found that the tendency was stronger in the low-amylose rice group.

As shown in Figure 2, the endo-cellulase activity of chalky unpolished rice grains of premium *Japonica* Koshihikari (*n* = 1) rice, ordinary *Japonica* rice (*n* = 7), and low-amylose *Japonica* rice (*n* = 2) were significantly lower than that of whole rice grains. The premium *Japonica* Koshihikari (*n* = 1) rice and Indica rice (*n* = 1) showed similarly tendencies. As a result, whole unpolished rice grains were shown to have significantly higher endo-cellulase activities than chalky unpolished rice grains, showing an opposite trend to the endo-xylanase activities. The whole unpolished rice grains showed markedly higher endo-cellulase activities, by 0.8–1.6 times, than chalky unpolished rice grains in 12 *Japonica* rice variants in 2021.

As a result of the statistical treatments using all the rice samples, the xylanase activities of chalky rice grains were shown to be higher than those of whole rice grains (*p* < 0.01) by Tukey’s one-way ANOVA. On the contrary, the cellulase activity of the chalky grains was shown to be lower than that of the whole grains (*p* < 0.05).

It seems very interesting that the activities of not only starch-related enzymes, but also cell wall-degrading enzyme activities, change along with the high-temperature ripening of rice grains.

Tsujii et al. reported that endogenous poly-galacturonase activities showed a positive correlation with pectin contents and a negative correlation with the hardness of cooked rice [13]. In our previous paper, xylanase activity showed positive correlations with amylose content and cellulase activity and a negative correlation with the adhesion of cooked rice [17]. Our results, shown in Figure 1 and Figure 2, are harmonized with the report about the discriminative DNA markers encoding 1,4-beta-xylanase and endo-1,4-beta-glucanase 13 in Indica rice, Indica–*Japonica* hybrid rice, and Javanica rice (tropical *Japonica* rice) [17].

Alpha-glucosidase hydrolyzes maltose and soluble starch to glucose, which is reported to affect the eating quality of rice grains [32]. Iwata et al. reported that alpha-glucosidase activity showed a positive correlation with GBSS activity and amylose content [33].

As shown in Appendix A, the α-amylase activity levels of chalky unpolished rice grains of premium *Japonica* Koshihikari (*n* = 1) rice, ordinary *Japonica* rice (*n* = 7), low-amylose *Japonica* rice (*n* = 2), and high-amylose *Japonica–Indica* hybrid rice (*n* = 1) were significantly higher than those of whole rice grains. These results were consistent with the report by Mitsui et al. [4] and our former reports [9,12].

As shown in Appendix A, the β-amylase activity levels of chalky unpolished rice grains of premium *Japonica* Koshihikari (*n* = 1) rice, ordinary *Japonica* rice (*n* = 3), low-amylose *Japonica* rice (*n* = 1), and high-amylose *Japonica-Indica* hybrid rice (*n* = 1) were significantly higher than those of whole rice grains. This result is in accordance with our previous report [9,12].

As shown in Appendix A, the protease activity of chalky unpolished grains of ordinary *Japonica* rice (*n* = 2), low-amylose *Japonica* rice (*n* = 1), and high-amylose *Japonica–Indica* hybrid rice (*n* = 1) were significantly higher than those of the whole rice grains.

As a result of statistical treatment using all the rice samples, the α-amylase and proteinase of the chalky grains showed significant higher activity levels than those of whole grains (*p* < 0.01). Unfortunately, beta-amylase activity did not show a significant difference between the two rice groups.

Our results are consistent with our previous report that the activity levels of not only α-amylase, but also β-amylase and protease, were higher for chalky rice grains [9,12,34]. Sun et al. reported that isoamylase is a factor in grain chalkiness using QTLs studies [35].

It has been reported that the gene expression of gibberellin is closely related with the activation of α-amylase, protease activity, and cell wall-degrading enzymes [36].

Our results may show that the high-temperature ripening has close relationship with the changes in the various enzyme activities affected by plant hormones, such as gibberellin [37].

The degradation of the cell walls of cereal grains during germination has been studied from a physiological viewpoint [25,38,39,40].

### 3.2. Activities of Various Hydrolytic Enzymes of Rice Grains Soaking in Hard Water

In our previous paper, we reported the different properties of whole grains and chalky grains. These chalky rice grains are characterized by high α- and *β*-amylase activity levels, high protease activity, low apparent amylose content, and low degrees of hardness and stickiness of boiled rice grains compared to whole grains [9]. In this study, we attempted to improve the physical properties of boiled chalky rice grains by reducing various enzyme activity levels.

As shown in Figure 3 and Appendix A, the α-amylase activity levels of whole grains soaked in purified water (0.05 ± 0.0 Ug^−1^) were significantly higher than those soaked in Evian (hardness: 304 mg/L, Ca: 8.0 mg, Mg: 2.6 mg, pH: 7.2) (0.03 ± 0.0 Ug^−1^) or Contrex (hardness: 1468 mg/L, Ca: 46.8 mg, Mg: 7.45 mg, pH: 7.2) (0.04 ± 0.0 Ug^−1^). On the contrary, those of two- or fourfold-diluted Evian and Contrex did not show inhibition of α-amylase activities.

As shown in Figure 3, the *β*-amylase activity levels of whole grains soaked in purified water (0.967 ± 0.0 Ug^−1^) were significantly higher than those soaked in Evian (0.630 ± 0.0 Ug^−1^), two- or fourfold dilutions of Evian (0.830 ± 0.0 Ug^−1^), (0.543 ± 0.0 Ug^−1^), Contrex (0.555 ± 0.0 Ug^−1^), or two- or fourfold dilutions of Contrex (0.612 ± 0.0 Ug^−1^), (0.648 ± 0.0 Ug^−1^).

As shown in Figure 3, the proteinase activity levels of the whole grains soaking in purified water (0.072 ± 0.0 Ug^−1^) were significantly higher than those soaking in Evian (0.065 ± 0.0 Ug^−1^), fourfold dilutions of Evian (0.066 ± 0.0 Ug^−1^), or fourfold dilutions of Contrex (0.066 ± 0.0 Ug^−1^). Those soaked in twofold dilutions of Evian (0.072 ± 0.0 Ug^−1^), Contrex (0.071 ± 0.0 Ug^−1^), or twofold dilutions of Contrex (0.072 ± 0.0 Ug^−1^) showed almost no reduction in proteinase activity.

As shown in Figure 3, the endo-xylanase activity levels of whole grains soaking in purified water (0.063 ± 0.0 Ug^−1^) were significantly lower than those of the samples soaking in Evian (0.077 ± 0.0 Ug^−1^). However, those soaked in Contrex (0.071 ± 0.0 Ug^−1^), two- or fourfold dilutions of Contrex (0.074 ± 0.0 Ug^−1^), (0.070 ± 0.0 Ug^−1^), and two- or fourfold dilutions of Evian (0.064 ± 0.0 Ug^−1^), (0.066 ± 0.0 Ug^−1^) did not show any significant effects regarding endo-xylanase activity.

As shown in Figure 3, the endo-cellulase activities of whole grains soaked in purified water (0.081 ± 0.0 Ug^−1^) were significantly lower than those soaking in two- or fourfold dilutions of Evian (0.138 ± 0.0 Ug^−1^), (0.095 ± 0.0 Ug^−1^), or fourfold dilutions of Contrex (0.110 ± 0.0 Ug^−1^). Those of the samples soaked in Evian (0.069 ± 0.0 Ug^−1^) were lower than those in the purified water group, and those soaked in Contrex (0.083 ± 0.0 Ug^−1^) and twofold dilutions of Contrex (0.078 ± 0.0 Ug^−1^) showed similar values to those in the purified water group.

As a result, various enzyme activities were inhibited by soaking in hard water with optimal concentrations. The activity levels of various hydrolytic enzymes in polished rice soaked in hard water showed a similar tendency to those of unpolished rice grains.

### 3.3. Textural Properties of Boiled Rice Grains

In our previous report, boiled chalky rice grains showed lower hardness and stickiness values and higher retrogradation degrees after boiling compared with the whole grains [12].

In the recent commercial *Japonica* rice market, rice grains containing about 30% of chalky rice are graded as low-class and priced lower than whole rice grains.

In this study, we measured the physical properties of boiled rice of whole and chalky grains after soaking in Evian, Contrex, and purified water using the individual grain method. Both low-compression (25%) and high-compression (90%) tests were conducted using a Tensipresser.

Ogawa et al. [41] reported that water absorption and the swelling of boiled rice, adding calcium, were inhibited compared to soaked in water.

As shown in Appendix A, H1 (the hardness of the surface layer of the boiled rice grains) and H2 (the hardness of the overall layer) of chalky boiled rice grains were significantly lower than those of whole boiled rice grains, a similarly tendency to that observed in our previous report [9,12]. Compared with other measurements, such as assays of enzyme activities (Appendix A, Table 1), the measurements of physical properties always show larger standard deviations, as shown in our previous report (references No. 9, No. 12, and No. 34).

As shown in Appendix A, the textural properties of chalky grains boiled after soaking in purified water change compared with whole grains, such as through an increase in hardness and stickiness.

In polished Koshihikari, H2 (hardness of the overall layer) and S1 (stickiness of the surface layer) of chalky boiled rice grains were significantly higher after soaking in Contrex compared to those of whole rice grains soaking in purified water, and the S2 (stickiness of overall layer) and L3 (the adhered of surface layer) of chalky boiled rice grains showed similar tendencies.

On the other hand, the S1 of chalky boiled rice grains after soaking in Evian was significantly higher compared to the whole rice grains soaked in purified water, while the H2, S2, and L3 of chalky boiled rice grains were slightly lower than those of whole rice grains soaked in purified water.

In polished Tsubusuke, the H2 and S1 of the chalky boiled rice grains after soaking in Contrex were significantly higher than those of whole rice grains soaked in purified water, and the S2 of chalky boiled rice grains showed a similar tendency. On the other hand, the S1 and S2 of chalky boiled rice grains after soaking in Evian were significantly higher than those of whole rice grains soaked in purified water, and the L3 of chalky boiled rice grains showed a similar tendency.

As shown in Table 1 and Appendix A, the ratio of the hardness of boiled 30% chalky blended rice after soaking in Evian to the hardness of the rice soaked in purified water was 1.05 ± 0.24 times (*n* = 5). The ratio of toughness was 1.04 ± 0.17 times (*n* = 6), the ratio of adhesion was 0.97 ± 0.14 times (*n* = 5), the ratio of stickiness was 1.06 ± 0.35 times (*n* = 5), and the ratio of cohesiveness was 1.05 ± 0.04 times (*n* = 8). The ratio of the hardness of those soaked in Contrex to the hardness of those soaked in purified water was 1.11 ± 0.23 times (*n* = 5), the ratio of toughness was 1.04 ± 0.19 times (*n* = 6), the ratio of adhesion was 1.06 ± 0.10 times (*n* = 7), the ratio of stickiness was 1.21 ± 0.44 times (*n* = 6), and cohesiveness was 1.05 ± 0.04 times (*n* = 9).

As a result, the various physical properties of boiled 30% chalky blended rice after soaking in hard water showed higher values than those of the rice soaked in purified water, and the physical properties of the rice soaked in Contrex were slightly higher than those of the rice soaked in Evian.

In the recent commercial market, rice grains containing about 30% chalky grains were classified as low-grade; thus, the prices are lower than those of whole rice grains. In our previous report, the hardness and toughness of boiled 30% chalky blended rice were lower than those of whole boiled rice grains, and the stickiness and adhesion showed a similar tendency.

We found that the boiled 30% chalky grain blended rice, after soaking in hard water, showed slightly higher hardness, toughness, stickiness, and cohesiveness compared to the rice soaked in purified water, which means that the physical properties of boiled 30% chalky blended rice were improved in terms of textural qualities.

### 3.4. d-Glucose, Maltose, and Saccharose Contents in Boiled Rice Grains

Awazuhara et al. [42] showed that the thermal dependency and stability of enzymes producing reducing sugar are different between outer endosperm and inner endosperm of rice.

As shown in Appendix A, the d-glucose, maltose, and saccharose in chalky boiled rice grains were significantly higher than those in whole boiled rice grains, which was consistent with our previous reports [9,34].

In Koshihikari, the d-glucose and maltose levels in chalky boiled rice grains after soaking in Evian and Contrex were significantly lower than those in grains soaked in purified water. The sugar content showed similarity to that of whole grains soaking in purified water. The saccharose contents of chalky boiled rice grains after soaking in Evian and Contrex showed a similar tendency.

In Tsubusuke, the d-glucose, maltose, and saccharose levels of chalky boiled rice grains after soaking in Evian and Contrex were lower than those soaked in purified water, and the sugar content was higher than that of whole grains soaked in purified water.

Shibuya et al. [31] demonstrated the isolation of cell walls from different parts of rice grains of a *Japonica* variety, their macromolecule composition, and the sugar linkages contained in these cell walls. Tsujii et al. [16] showed that the extent of decomposition of pectin has a negative correlation with the hardness value of cooked rice.

In two kinds of boiled rice, after soaking in Evian, Contrex, or purified water, the stickiness of the overall layer (S2) of boiled rice showed a positive correlation with d-glucose (r = 0.70, *p* < 0.05), maltose (r = 0.77, *p* < 0.01), and saccharose (r = 0.78, *p* < 0.01). The hardness of the surface layer (H1) of boiled rice showed a negative correlation with d-glucose (r = −0.74, *p* < 0.01), maltose (r = −0.68, *p* < 0.05), and saccharose (r = −0.76, *p* < 0.01), and the endo-cellulase activities showed a negative correlation with d-glucose (r = −0.67, *p* < 0.05), maltose (r = −0.58, *p* < 0.05), and saccharose (r = −0.58, *p* < 0.05). Furthermore, α-amylase activity showed a negative correlation with the stickiness of surface layer (S1) (r = −0.70, *p* < 0.05) of boiled rice, as shown in Appendix A.

It was shown that rice grains boiled after soaking in Evian or Contrex contained lower amounts of glucose, due to the lower enzyme activity, than of the rice grains boiled after soaking in purified water.

As shown in Table 2, the d-glucose content of the 30% chalky blended boiled grains after soaking in Evian or Contrex was significantly lower than that of blended rice soaked in purified water (*n* = 10), and the maltose and saccharose contents showed similar tendencies.

The ratio of d-glucose of the boiled 30% chalky blended rice after soaking in Evian to d-glucose of the rice soaking in purified water was 0.82 ± 0.05 times (*n* = 10). The ratio of maltose was 0.86 ± 0.08 times (*n* = 8), that of saccharose was 0.92 ± 0.04 times (*n* = 10), and of the results for blended boiled grains after soaking in Contrex showed a similar tendency. As a result, the sugar content of the boiled 30% chalky blended rice were lower after soaking in hard water compared to that of the boiled 30% chalky blended rice soaked in purified water.

In our previous report, the sugar contents of the boiled 30% chalky blended rice after soaking in purified water were 1.1 times higher than those of 100% whole grain rice [9]. The reason why blended boiled rice contained a higher sugar content than 100% whole grains is due to the higher activity levels of multiple amylases and lower activity levels of starch-synthesizing enzymes.

In this study, it was shown that the boiled 30% chalky blended rice, after soaking in Evian and Contrex, had a lower sugar content than that of the rice boiled after soaking in purified water due to lower enzyme activity.

Onishi et al. [43] showed that cooked rice boiled in hard water is harder, with less coloring than rice boiled in soft water.

As shown in Appendix A, a ratio of the color difference (ΔE*(ab)) of boiled 30% chalky blended rice after soaking in Evian to the color difference of the rice soaked in purified water was 1.06 ± 0.19 times, and those of a ratio of the color difference of rice soaked in Contrex was 0.92 ± 0.18 times.

As a result, the color difference of boiled 30% chalky blended rice after soaking in Evian showed slightly higher values than of the rice soaked in purified water, while the color differences in the rice soaked in Contrex were slightly lower.

In this report, the color difference of boiled 30% chalky blended rice after soaking in hard water showed a similar tendency to the boiled rice after soaking in purified water. Although it is well-known that color differences in boiled rice are affected by amino-carbonyl reactions from sugar and amino acids, in our investigation, various enzyme activities were inhibited by soaking in hard water of an optimal concentration. The reason why blended 30% chalky rice, when boiled after soaking in Evian or and Contrex, did not show a marked color difference was due to the lower sugar and amino acid contents than the rice boiled after soaking in purified water. As multiple amylases and proteinase activity levels were reduced by the hard water, the boiled rice contained lower amounts of mono-or oligosaccharides, as shown in Appendix A.

### 3.5. Calcium and Magnesium Contents in Whole and Chalky Polished or Unpolished Rice, and Those in Boiled Rice, after Soaking

The ash distribution in brown rice is calculated as 51% in bran, 10% in germ, 11% in polish, and 28% in milled rice. Some minerals are also present according to some calculations, which have shown that milled rice retains 63% of the sodium and 74% of the calcium content of brown rice [44].

Calcium deficiency is a global problem, especially in the aging population [45]. Significant impairments in bone mineral density and bone fracture are generally found in low-calcium-intake populations [46]. To supplement calcium deficiency, boiled rice as a staple food is one of the best ways, because many people eat rice almost every day.

As shown in Table 3, unpolished chalky rice contained about 1.3 times more calcium than unpolished whole rice. Similarly, polished chalky rice contained about 1.3 times more calcium than whole rice in the case of Koshihikari.

As a result, it seems that the calcium levels in chalky unpolished or polished rice grains were significantly higher than in whole rice grains, and magnesium showed a similar tendency. Okuda showed that minerals and protein contents, negatively influence the quality of sake in abundance, are distributed more in the outer layer [20,21].

Polished whole rice grains boiled after soaking in Evian contained 3.5 times more calcium than those boiled after soaking in purified water, and furthermore, those boiled after soaking in Contrex contained as much as 16.5 times more than those boiled after soaking in purified water in the case of Koshihikari.

In the case of Tsubusuke, chalky polished rice grains boiled after soaking in Evian or Contrex contained 2.7 times and 13.5 times more calcium than those boiled after soaking in purified water. Additionally, whole polished rice grains boiled after soaking in Evian or Contrex contained 2.6 times and 13.5 times more calcium than those boiled after soaking in purified water.

As shown in Table 3, in polished Koshihikari rice, the magnesium content in whole and chalky grains showed a similar tendency with calcium, and polished chalky Koshihikari rice grains showed a slightly higher magnesium content than whole grains.

Polished whole rice grains boiled after soaking in Evian contained 1.3 times more magnesium than rice boiled after soaking in purified water, and furthermore, rice boiled after soaking in Contrex contained about 1.8 times more than rice boiled after soaking in purified water in the case of Koshihikari.

In polished whole or chalky boiled Tsubusuke rice grains, the ratio of magnesium content in chalky rice boiled after soaking in Evian or Contrex to rice boiled after soaking in purified water was 1.1 or 1.3. The ratio of magnesium content in whole rice boiled after soaking in Evian or Contrex to rice boiled after soaking in purified water was 1.1 or 1.6.

As a result, it became possible to improve the quality of the chalky rice grains not only in terms of textural properties, but also in terms of bio-functionality and mineral absorption, by boiling in hard water. According to the Dietary Reference Intakes for Japanese (2020), it is recommended to absorb 650 mg of calcium and 370 mg of magnesium per day. If one eats 300 g of boiled rice soaked in Contrex every day, he or she can ingest 342 mg of calcium and 110 mg of magnesium. These figures mean that about 53% of the calcium and 30% of the magnesium necessary per day can be consumed from only boiled rice.

## 4. Conclusions

In the present paper, we investigated the characteristics of chalky rice grains generated by ripening under high temperatures, compared with whole grains. We found that the activity levels of not only endogenous amylase and proteinase, but also cell wall-degrading enzymes such as xylanase and cellulase, changed markedly. As we reported in the former study boiled grains of chalky rice become softer and non-stickier compared with boiled grains of whole rice, it was ascertained that the change was intrinsic for the chalky rice grains. It was shown that xylanase, in addition to amylase and proteinase, may take on an important role contributing to the change in texture of the boiled chalky rice grains. In order to prevent the above-mentioned deterioration in the texture of the boiled chalky rice grains, we used hard water, such as Evian or Contrex, to soak and cook the chalky rice grains. It was shown that the hard water was useful for prevention of the texture deterioration of the boiled rice grains due to inhibition of the activities of endogenous hydrolytic enzymes, such as α-amylase, β-amylase, and proteinase. Furthermore, we found that hard water was useful for a 2.6- to 16.5-fold increase in calcium absorption through consumption of boiled rice grains soaked and cooked using hard water.

## Figures and Tables

**Figure 1 foods-12-02510-f001:**
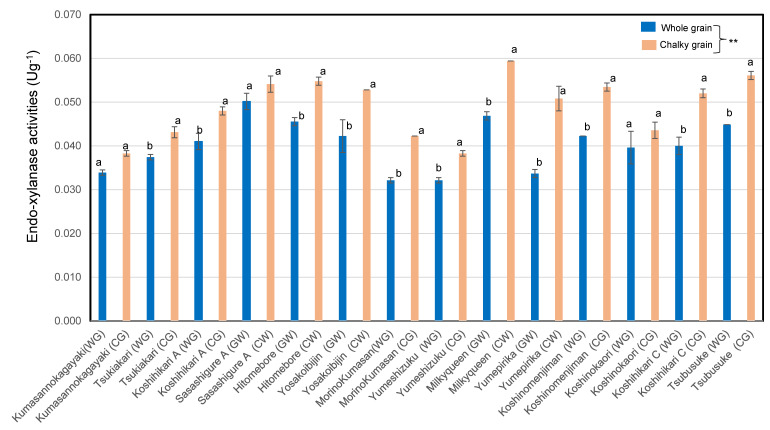
End-xylanase activities of whole and chalky rice grains of 14 *Japonica* unpolished rice samples in 2021. Different letters (a, b) indicate that whole and chalky grains of each same rice samples are significantly different. ** Correlation is significant at 1% according to the method of Tukey’s multiple comparison.

**Figure 2 foods-12-02510-f002:**
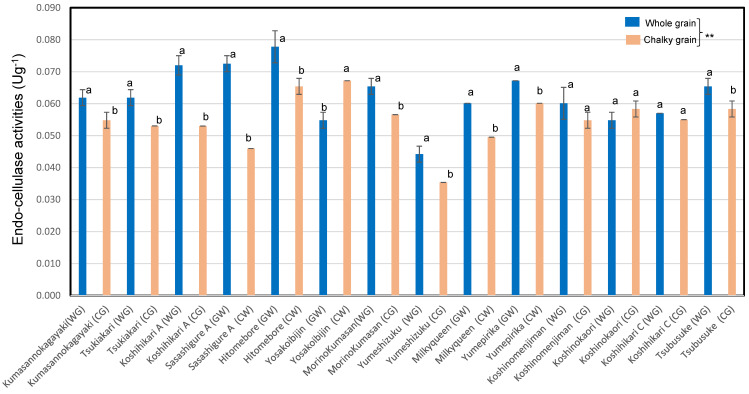
Cellulase activities of whole and chalky rice grains in 14 unpolished *Japonica* rice samples in 2021. Different letters (a, b) indicate that whole and chalky grains in each rice sample are significantly different. ** Correlation is significant at 1% according to the method of Tukey’s multiple comparison.

**Figure 3 foods-12-02510-f003:**
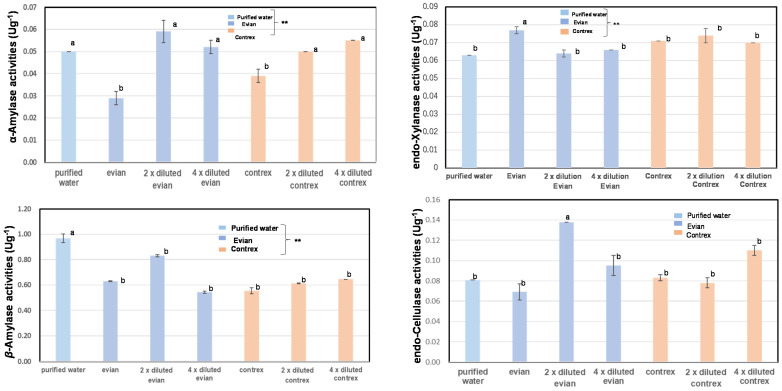
α- and β-amylase activity, proteinase activity, and cell wall-degrading enzyme activity of unpolished rice soaking in Evian, Contrex, or purified water. Within each measurement (α-amylase, β-amylase, proteinase, endo-xylanase, endo-cellulase) in the same column, different letters (a, b) indicate that whole unpolished Koshihikari rice flour, when soaked in Evian, two- or fourfold dilutions of Evian, Contrex, two- or fourfold dilutions of Contrex, or purified water were significantly different. Different letters (a, b) indicate that samples soaked in different types of water were significantly different. ** Correlation was significant at 1% according to the method of Tukey’s multiple comparison.

**Table 1 foods-12-02510-t001:** Physical properties of in 10 kinds of boiled *Japonica* rice with 30% chalky grains, blended after soaking in Evian, Contrex, and purified water, in 2021.

	Hardness	Toughness	Adhesion	Stickiness	Cohesiveness
	×10^5^ (N/cm^2^)	×10^5^ (N/cm^2^)	×10^5^ (N/cm^2^)	×10^5^ (N/cm^2^)	(A6/A5)
Kumasannokagayaki (purified water)	0.86 ± 0.05 b	17.12 ± 0.53 a	12.41 ± 0.49 b	12.24 ± 1.33 b	0.38 ± 0.01 a
Kumasannokagayaki (Evian)	1.15 ± 0.17 a	17.96 ± 0.46 a	12.67 ± 2.35 b	17.79 ± 5.89 b	0.38 ± 0.02 a
Kumasannokagayaki (Contrex)	1.38 ± 0.05 a	18.33 ± 0.87 a	14.40 ± 1.16 a	26.70 ± 079 a	0.38 ± 0.00 a
Tsukiakari (purified water)	1.63 ± 0.03 a	18.03 ± 0.71 a	12.44 ± 2.02 b	17.94 ± 1.78 a	0.38 ± 0.02 a
Tsukiakari (Evian)	1.23 ± 0.18 b	19.96 ± 0.80 a	12.64 ± 1.20 b	19.13 ± 4.21 a	0.41 ± 0.01 a
Tsukiakari (Contrex)	1.56 ± 0.37 a	19.22 ± 2.47 a	13.48 ± 0.00 a	17.55 ± 1.00 a	0.40 ± 0.00 a
Koshihikari A (purified water)	1.14 ± 0.19 b	16.16 ± 0.41 a	14.51 ± 0.29 a	25.66 ± 1.34 a	0.38 ± 0.00 a
Koshihikari A (Evian)	1.67 ± 0.19 a	17.22 ± 2.56 a	12.33 ± 1.12 a	19.74 ± 4.50 b	0.41 ± 0.04 a
Koshihikari A (Contrex)	1.29 ± 0.09 b	16.89 ± 1.06 a	14.16 ± 1.05 a	25.95 ± 4.67 a	0.41 ± 0.05 a
Sasashigure A (purified water)	1.36 ± 0.13 b	13.32 ± 0.74 b	12.83 ± 1.49 b	22.05 ± 5.50 b	0.41 ± 0.05 a
Sasashigure A (Evian)	1.28 ± 0.46 b	19.35 ± 1.47 a	13.52 ± 2.37 b	24.23 ± 1.52 b	0.42 ± 0.02 a
Sasashigure A (Contrex)	1.70 ± 0.11 a	19.72 ± 1.63 a	15.30 ± 1.66 a	31.40 ± 3.19 a	0.42 ± 0.01 a
Hitomebore (purified water)	1.59 ± 0.25 a	19.92 ± 2.29 a	15.02 ± 0.28 a	22.91 ± 0.63 a	0.37 ± 0.03 a
Hitomebore (Evian)	1.23 ± 0.07 b	17.55 ± 1.25 b	12.78 ± 0.19 b	14.80 ± 0.68 b	0.38 ± 0.00 a
Hitomebore (Contrex)	1.54 ± 0.06 a	17.88 ± 1.79 b	13.35 ± 0.45 b	24.13 ± 6.77 a	0.41 ± 0.02 a
Yosakoibijin (purified water)	1.28 ± 0.10 a	22.07 ± 2.55 a	10.65 ± 0.14 a	16.04 ± 5.47 a	0.39 ± 0.05 a
Yosakoibijin (Evian)	1.40 ± 0.42 a	20.11 ± 3.03 a	13.87 ± 1.56 a	21.20 ± 0.16 a	0.42 ± 0.04 a
Yosakoibijin (Contrex)	1.27 ± 0.46 a	18.17 ± 2.55 b	11.21 ± 0.84 a	18.97 ± 4.47 a	0.40 ± 0.04 a
Koshihikari B (purified water)	1.66 ± 0.27 a	21.43 ± 0.29 a	12.80 ± 1.10 b	30.99 ± 0.74 a	0.39 ± 0.05 a
Koshihikari B (Evian)	1.38 ± 0.11 a	18.65 ± 0.47 b	12.33 ± 1.96 b	23.93 ± 0.15 b	0.42 ± 0.04 a
Koshihikari B (Contrex)	1.42 ± 0.22 a	18.69 ± 1.04 b	15.18 ± 0.96 a	22.87 ± 6.17 b	0.40 ± 0.04 a
Sasashigure B (purified water)	1.46 ± 0.40 b	19.56 ± 1.07 b	11.09 ± 0.26 a	17.89 ± 0.84 a	0.39 ± 0.03 a
Sasashigure B (Evian)	1.45 ± 0.21 b	22.26 ± 0.92 a	11.01 ±0.07 a	17.66 ± 0.24 a	0.40 ± 0.04 a
Sasashigure B (Contrex)	1.88 ± 0.01 a	22.76 ± 0.82 a	11.61 ± 0.31 a	16.70 ± 4.54 a	0.41 ± 0.01 a
Morinokumasan (purified water)	0.92 ± 0.00 a	17.84 ± 0.11 b	14.68 ± 0.33 a	28.51 ± 3.76 a	0.40 ± 0.00 a
Morinokumasan (Evian)	1.16 ± 0.34 a	18.49 ± 2.41 a	12.86 ± 0.52 a	21.81 ± 2.96 b	0.40 ± 0.02 a
Morinokumasan (Contrex)	1.08 ± 0.16 a	17.80 ± 1.67 b	14.38 ± 0.83 a	25.61 ± 1.72 b	0.41 ± 0.00 a
Yumeshizuku (purified water)	1.35 ± 0.25 b	18.90 ± 1.40 a	15.12 ± 1.85 a	18.48 ± 5.02 b	0.37 ± 0.05 b
Yumeshizuku (Evian)	1.47 ± 0.21 b	17.68 ± 1.14 a	12.20 ± 1.87 b	31.78 ± 2.58 a	0.42 ± 0.01 a
Yumeshizuku (Contrex)	1.23 ± 0.01 a	18.69 ± 1.85 a	15.14 ± 0.83 a	31.86 ± 0.85 a	0.42 ± 0.04 a

The physical properties of boiled grains of blended rice were measured using bulk measurements, which included low compression (compression ratio = 23%: twice), intermediate compression (compression ratio = 46%: twice), and high compression (compression ratio = 92%: twice), according to the 2 × 3 bite method. Within each measurement (hardness, toughness, adhesion, stickiness, cohesiveness) in the same column, different letters (a, b) mean that blended boiled rice in each rice sample was significantly different. The value of hardness is indicated by the height and that of toughness by the area of continuous progressive compression in Tensipresser. Cohesiveness (A6/A5) is shown as the ratio of hardness under high compression. Values are shown as mean ± standard deviation.

**Table 2 foods-12-02510-t002:** Oligosaccharides of boiled 30% chalky blended rice after soaking in Evian, Contrex, and purified water in 10 kinds of *Japonica* rice in 2021.

	d-Glucose Content	Maltose Content	Saccharose Content
	(g/100 g)	(g/100 g)	(g/100 g)
Kumasannokagayaki (purified water)	0.065 ± 0.004 a	0.095 ± 0.002 a	0.313 ± 0.008 a
Kumasannokagayaki (evian)	0.058 ± 0.001 b	0.091 ± 0.001 a	0.299 ± 0.001 a
Kumasannokagayaki (contrex)	0.055 ± 0.002 b	0.087 ± 0.002 a	0.299 ± 0.000 a
Tsukiakari (purified water)	0.078 ± 0.002 a	0.134 ± 0.001 a	0.302 ± 0.002 a
Tsukiakari (evian)	0.057 ± 0.002 b	0.090 ± 0.000 b	0.260 ± 0.006 b
Tsukiakari (contrex)	0.055 ± 0.002 b	0.088 ± 0.007 b	0.254 ± 0.007 b
Koshihikari A (purified water)	0.074 ± 0.001 a	0.100 ± 0.000 a	0.345 ± 0.000 a
Koshihikari A (evian)	0.058 ± 0.002 b	0.089 ± 0.000 b	0.311 ± 0.006 b
Koshihikari A (contrex)	0.057 ± 0.001 b	0.089 ± 0.001 b	0.306 ± 0.008 b
Sasashigure A (purified water)	0.077 ± 0.002 a	0.097 ± 0.000 a	0.331 ± 0.009 a
Sasashigure A (evian)	0.063 ± 0.002 b	0.087 ± 0.001 b	0.313 ± 0.006 a
Sasashigure A (contrex)	0.066 ± 0.002 b	0.091 ± 0.002 a	0.322 ± 0.009 a
Hitomebore (purified water)	0.062 ± 0.001 a	0.088 ± 0.001 a	0.281 ± 0.008 a
Hitomebore (evian)	0.050 ± 0.001 b	0.073 ± 0.003 b	0.259 ± 0.007 a
Hitomebore (contrex)	0.046 ± 0.001 b	0.072 ± 0.002 b	0.252 ± 0.006 a
Yosakoibijin (purified water)	0.060 ± 0.001 a	0.115 ± 0.003 a	0.514 ± 0.007 a
Yosakoibijin (evian)	0.051 ± 0.001 b	0.107 ± 0.001 a	0.492 ± 0.002 a
Yosakoibijin (contrex)	0.055 ± 0.001 b	0.112 ± 0.001 a	0.500 ± 0.005 a
Koshihikari B (purified water)	0.070 ± 0.001 a	0.092 ± 0.001 a	0.295 ± 0.001 a
Koshihikari B (evian)	0.061 ± 0.001 b	0.083 ± 0.001 b	0.280 ± 0.003 a
Koshihikari B (contrex)	0.063 ± 0.001 b	0.082 ± 0.001 b	0.277 ± 0.001 a
Sasashigure B (purified water)	0.062 ± 0.000 a	0.085 ± 0.000 a	0.295 ± 0.000 a
Sasashigure B (evian)	0.051 ± 0.000 b	0.073 ± 0.003 b	0.270 ± 0.001 a
Sasashigure B (contrex)	0.054 ± 0.002 b	0.073 ± 0.001 b	0.277 ± 0.001 a
Morinokumasan (purified water)	0.075 ± 0.000 a	0.113 ± 0.003 a	0.411 ± 0.001 a
Morinokumasan (evian)	0.063 ± 0.001 b	0.099 ± 0.003 a	0.378 ± 0.003 a
Morinokumasan (contrex)	0.066 ± 0.000 b	0.104 ± 0.001 a	0.385 ± 0.001 a
Yumeshizuku (purified water)	0.074 ± 0.000 a	0.098 ± 0.001 a	0.318 ± 0.002 a
Yumeshizuku (evian)	0.057 ± 0.001 b	0.079 ± 0.001 b	0.275 ± 0.001 b
Yumeshizuku (contrex)	0.060 ± 0.000 b	0.084 ± 0.001 b	0.285 ± 0.001 b

Within each measure (d-glucose content, maltose content, saccharose content) in the same column, different letters (a, b) mean that in each sample, the properties of the rice were significantly different. Values are shown as mean ± standard deviation.

**Table 3 foods-12-02510-t003:** Calcium and magnesium contents of whole and chalky unpolished rice, those of polished rice, and those of polished rice boiled after soaking in Evian, Contrex, and purified water using kinds of *Japonica* rice in 2021.

	Calcium	Magnesium
	(mg/100 g)	(mg/100 g)
Koshihikari C (WG) (unpolished rice)	9.6 ± 0.0 b	103.0 ± 0.0 a
Koshihikari C (CG) (unpolished rice)	12.3 ± 0.0 a	102.0 ± 0.0 a
Koshihikari C (WG) (polished rice)	5.4 ± 0.0 b	19.3 ± 0.0 b
Koshihikari C (CG) (polished rice)	6.8 ± 0.0 a	22.5 ± 0.0 a
Koshihikari C (WG) (polished boiled rice) (pulified water)	5.5 ± 0.0 b	17.5 ± 0.0 b
Koshihikari C (CG) (polished boiled rice) (pulified water)	6.9 ± 0.0 a	20.4 ± 0.0 a
Koshihikari C (WG) (polished boild rice) (Evian)	19.0 ± 0.0 b	22.1 ± 0.0 b
Koshihikari C (CG) (polished boild rice) (Evian)	23.9 ± 0.0 a	25.8 ± 0.0 a
Koshihikari C (WG) (polished boiled rice) (Contrex)	90.5 ± 0.0 b	31.3 ± 0.0 b
Koshihikari C (CG) (polished boiled rice) (Contrex)	114.0 ± 0.0 a	36.5 ± 0.0 a
Tsubusuke (WG) (polished boiled rice) (pulified water)	6.1 ± 0.0 a	26.6 ± 0.1 b
Tsubusuke (CG) (polished boiled rice) (pulified water)	6.4 ± 0.0 a	29.4 ± 0.1 a
Tsubusuke (WG) (polished boiled rice) (Evian)	15.8 ± 0.0 b	30.4 ± 0.1 a
Tsubusuke (CG) (polished boiled rice) (Evian)	17.5 ± 0.0 a	32.6 ± 0.1 a
Tsubusuke (WG) (polished boiled rice) (Contrex)	82.6 ± 0.2 b	41.3 ± 0.1 a
Tsubusuke (CG) (polished boiled rice) (Contrex)	86.1 ± 0.2 a	39.0 ± 0.1 a

Within each measure (calcium, magnesium) in the same column, different letters (a, b) mean that each rice sample was significantly different. Abbreviation: WG, whole grains; CG, chalky grains. Values are shown as mean ± standard deviation.

## Data Availability

The datasets generated for this study are available upon request to the corresponding author.

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
