# Peer review of "Effects of Hard Water Boiling on Chalky Rice in Terms of Texture Improvement and Ca Fortification"

_foods, 2023, doi:10.3390/foods12132510_

Round 1
Reviewer 1 Report
Authors investigated the effects of hard water boiling on textural properties and Ca fortification of chalky. The work is a novel study, but there are major concerns which need to be addressed.
Line 251: “As shown in Figure 1, the endo-xylanase activities of chalky unpolished rice grains of premium Japonica rice Koshihikari (n=2), ordinary Japonica rice (n=6), low-amylose Japonica rice (n=2) and high-amylose Japonica-Indica hybrid rice (n=1) were significantly higher than those of whole rice grains.” How do you conclude this? You didn’t analyze all samples together. Firstly, analyze all samples using GLM, then decide if there is significant difference or not.
Do the same analysis for all studied traits in section 3.1.
Fig 1, 2 and Table 1: Authors compared two groups (WG vs CG) and put the letter on each column to show the significant difference between them. Its normal to show the higher mean with “a” and the lower with “b”. Authors used different letters for higher means. To compare two groups, it’s usual to use t-test instead of Tukey test.
Line 360-363: There is not significant difference between purified water and 2 or 4-fold dilution of Evian and Contrex. Also Evian and Contrex increased α-amylase activity.
Table 2: There is different response in term of using Evian and Contrex. Its recommended to analyze all samples to compare two hard water as well.
Table 3: Hardness (H2) column: higher means in Koshihikari received “b”, confusing.
For Tsubusuke: difference between Tsubusuke (WG) (pulified water) and Tsubusuke (CG) (Contrex) (1839-1716=123) is significant but difference between Tsubusuke (CG) (Contrex) and Tsubusuke (CG) (pulified water) (1716-1590=126) is higher but not significant. Please check all analysis.
Table 4: Re-analyze and check all significance. For example: Hardness for Yumeshizuku: difference between 1.47-1.35=12 is not significant but difference between 1.35-1.23=12 is significant.
Fig 3: according to the error bar difference between Evian and Contrex for stickiness isn’t significant.
Table 6 & 7: same problem with analysis and significance letter.
Author Response
Reviewer1
Comments and Suggestions for Authors
Authors investigated the effects of hard water boiling on textural properties and Ca fortification of chalky. The work is a novel study, but there are major concerns which need to be addressed.
A: We are very grateful for your kind and valuable comments. As we revised our manuscript according to your comments, we express our gratitude for your re-reviewing in advance.
- Line 251: “As shown in Figure 1, the endo-xylanase activities of chalky unpolished rice grains of premium Japonica rice Koshihikari (n=2), ordinary Japonica rice (n=6), low-amylose Japonica rice (n=2) and high-amylose Japonica-Indica hybrid rice (n=1) were significantly higher than those of whole rice grains.” How do you conclude this? You didn’t analyze all samples together. Firstly, analyze all samples using GLM, then decide if there is significant difference or not.
A: Thank you for your valuable comment. According to your comment, we revised.
As results of the statistical treatments using all the rice samples, xylanase activities of chalky rice grains were shown to be higher than those of whole rice grains (p < 0.01) by Tukey’s one-way analysis of ANOVA, on the contrary, cellulase activities of the chalky grains were shown to be lower than those of whole grains (p < 0.05). We added these results in L300-L303.
In the former manuscript, we described classifying various rice cultivars to compare each class of rice cultivars because we had found that the low-amylose rice tended to be suffered by high-temperature ripening more than the other rice cultivars in our previous report ([9]).
According to your comment, we added statistical treatment to compare the chalky rice grains and whole rice grains using all the rice samples in the statistical treatment.
- Do the same analysis for all studied traits in section 3.1.
A: As results of statistical treatment using all the rice samples, α-amylase and proteinase activities of the chalky grains showed significant higher activities than those of whole grains (p < 0.01). Unfortunately, beta-amylase activity did not show the significant difference between the two rice groups.
We added these results in L359-L362.
- Fig 1, 2 and Table 1: Authors compared two groups (WG vs CG) and put the letter on each column to show the significant difference between them. Its normal to show the higher mean with “a” and the lower with “b”. Authors used different letters for higher means. To compare two groups, it’s usual to use t-test instead of Tukey test.
A: Thank you for your comment. We revised Figure 1, 2 and Table 1, 2, 4 and all the Supplemental Tables. According to your comment, we used t-test for the comparison between the two groups.
- Line 360-363: There is not significant difference between purified water and 2 or 4-fold dilution of Evian and Contrex. Also Evian and Contrex increased α-amylase activity.
A: Thank you for your comment. As you point out, 2 or 4-fold dilution of Evian did not show the significant difference with purified water. We think that calcium in the hard water affects the enzyme activity. As calcium content of Evian is much lower than Contrex, 2 or 4-fold Evian does not influence, but 2 or 4-fold diluted Contrex influence due to higher calcium content than 2 or 4-fold diluted Evian.
- Table 2: There is different response in term of using Evian and Contrex. Its recommend
to analyze all samples to compare two hard water as well.
A: Thank you for your very valuable comment. After the comparison between two kinds of hard water, we revised to “on the contrary, those of 2 or 4-fold diluted Evian and Contrex did not show the inhibition of α-amylase activities.” (L384-L387) from “and those of 2 or 4-fold dilution of Evian and Contrex showed a similar tendency to reduce α-amylase activity”. For proteinase activity, we revised to “almost no reduction of” from “a similar tendency to reduce” proteinase activity (L396-L397).
For endo-xylanase, we revised to “As shown in Figure 3, endo-xylanase activities of whole grains soaking in purified water (0.063 ± 0.0 Ug-1) were significantly lower than those soaking in Evian (0.077 ± 0.0 Ug-1), but Contrex (0.071 ± 0.0 Ug-1) and 2 or 4- fold dilution of Contrex (0.074 ± 0.0 Ug-1), (0.070 ± 0.0 Ug-1), and those of 2 or 4- fold dilution of Evian (0.064 ± 0.0 Ug-1), (0.066 ± 0.0 Ug-1) did not showed a similar tendency to activate any significant effects to endo-xylanase activity.”(L398-L403) For cellulase, And for cellulase, we revised from “tendency” to “values” in L409.
And we changed Table 2 to supplement table 2 and made new figure 3 according to the comment by the other reviewer. We corrected the letter “a”, and “b” in supplement Table 2.
- Table 3: Hardness (H2) column: higher means in Koshihikari received “b”, confusing.
A: Thank you for your comment. We revised “a” and “b”in the Table.
- For Tsubusuke: difference between Tsubusuke (WG) (purified water) and Tsubusuke (CG) (Contrex) (1839-1716=123) is significant but difference between Tsubusuke (CG) (Contrex) and Tsubusuke (CG) (purified water) (1716-1590=126) is higher but not significant. Please check all analysis.
A: We are sorry for our mistake. We revised Table 3 and changed it to supplement table 3.
- Table 4: Re-analyze and check all significance. For example: Hardness for Yumeshizuku: difference between 1.47-1.35=12 is not significant but difference between 1.35-1.23=12 is significant.
A: Thank you for your comment. Although the differences are same values (0.12) between Hardness of Yumesizuku (purified water, Evian, Contrex) as you pointed out, standard deviations are 0.25, 0.21, and 0.01. Therefore, as a result of statistical treatment, only Contrex showed significant difference with other two samples.
- Fig 3: according to the error bar difference between Evian and Contrex for stickiness isn’t significant.
A: Thank you for your comment. As a result of statistical treatment (t-test), r for stickiness was 0.79 (p < 0.01) between Evian and Contrex.
- Table 6 & 7: same problem with analysis and significance letter.
A: Thank you for your valuable comment. We revised in Table 6 & 7.

Reviewer 2 Report
There's a lot of experimental data in the manuscript, but it's not well presented. There are some suggestions for improvement of the manuscript.
1. The novelty of the manuscript is unclear. Lines 63-78 should be rewritten.
2. The hardness and pH of purified water (treated tap water with a water purifier) should be provided.
3. There are some format errors in the manuscript (such as Lines 95, 99, 228, 238, 247, 295), please check the whole manuscript carefully.
4. There are too many paragraphs in 3. Results and Discussion. Some information could be merged.
5. The paper needs more discussion, Such as (3.2. Activities of various hydrolytic enzymes of rice grains soaking in hard water).
6. There are 7 tables in the manuscript, which contain a lot of data. But it's better to express it in an intuitive method (such as figures).
7. The conclusion should be more concise. Some key data could be presented in the conclusion.
8. Why use commercially available mineral water instead of water with a specific hardness?
Author Response
Reviewer2
Comments and Suggestions for Authors
There's a lot of experimental data in the manuscript, but it's not well presented. There are some suggestions for improvement of the manuscript.
- The novelty of the manuscript is unclear. Lines 63-78 should be rewritten.
A: Thank you for your comment. We added L63-L67, and L71-L81 to clarify the novelties of our report.
- The hardness and pH of purified water (treated tap water with a water purifier) should be provided.
A: Thank you for your comment. We added it in L.105-L106.
- There are some format errors in the manuscript (such as Lines 95, 99, 228, 238, 247, 295), please check the whole manuscript carefully.
A: Thank you for your comment. We checked and corrected the format for all the format.
- There are too many paragraphs in 3. Results and Discussion. Some information could be merged.
A: Thank you for your comment. According to your advice, we merged 3.8 and 3.9.
- The paper needs more discussion, Such as (3.2. Activities of various hydrolytic enzymes of rice grains soaking in hard water).
A: Thank you for your comment. We revised discussion as below;
As results of the statistical treatments using all the rice samples, xylanase activities of chalky rice grains were shown to be higher than those of whole rice grains (p < 0.01) by Tukey’s one-way analysis of ANOVA, on the contrary, cellulase activities of the chalky grains were shown to be lower than those of whole grains (p < 0.05). (L.300-303)
As results of statistical treatment using all the rice samples, α-amylase and pro-teinase activities of the chalky grains showed significant higher activities than those of whole grains (p < 0.01). Unfortunately, beta-amylase activity did not show the signifi-cant difference between the two rice groups. (L.359-L362)
on the contrary, those of 2 or 4-fold diluted Evian and Contrex did not show the inhi-bition of α-amylase activities. (L384-L387)
but Contrex (0.071 ± 0.0 Ug-1) and 2 or 4- fold dilution of Contrex (0.074 ± 0.0 Ug-1), (0.070 ± 0.0 Ug-1), and those of 2 or 4- fold dilution of Evian (0.064 ± 0.0 Ug-1), (0.066 ± 0.0 Ug-1) did not show any significant effects to endo-xylanase activity. (L.400-403)
Compared with the other measurements, such as assay of enzyme activities (supple-ment Table1, Table 1,), standard deviations of the measurements of physical proper-ties always show larger standard deviations as shown in our previous report (refer-ences No.9, No.12, and No.34). (L439-L442)
- There are 7 tables in the manuscript,which contain a lot of data. But it's better to express it in an intuitive method (such as figures).
A: Thank you for your comment. We changed table 1 to supplement table 1, table 2 to figure 3 and supplement table 2, table 3 to supplement table 3, figure 3 to supplement figure 1, table 5 to supplement table 3. According to your advice, we reduced the numbers of table and figure. As we used two methods for the textural measurements, bulk method and high-pressure/low pressure method, we moved the data (table 3) by the latter method to the supplemental table 2.
- The conclusion should be more concise. Some key data could be presented in the conclusion.
A: Thank you for your comment. We revised conclusion as below;
Global warming impairs grain filling in rice and leads to chalky-appearing grains, which were damaged in their physicochemical and cooking qualities . In the present paper, we investigated the characteristics of chalky rice grains, generated by ripening under high-temperature, comparing with whole grains. We found that not only activities of endogenous amylase and proteinase but also cell-wall degrading enzymes, such as xylanase and cellulase, change markedly. As we reported in the former report that boiled rice grains from chalky rice become softer and non-stickier compared with boiled rice grains from whole rice, it was ascertained the change is intrinsic for the chalky rice grains. It was shown that xylanase, in addition to amylase and proteinase, may take an important role to the change in texture of the boiled chalky rice grains.
In order to prevent above-mensioned deterioration in the texture of the boiled rice grains from chalky rice, we tried to use hard water, such as Evian or Contrex, for soaking and cooking of chalky rice garins. It was shown that the hard water is useful for the prevention of the texture deterioration of the boiled rice grains due to the inhibition of the reduction of endogenous hydrolytice enzymes, such as α-amylase, β-amylase amylase, and proteinase, and xylanase. Furthermore, we found that hard water is useful for the 2.6 to 16.5 times increase of calcium absorptuin through the meal with the boiled rice grains soaked and cooked using hard water.
- Why use commercially available mineral water instead of water with a specific hardness?
A: Thank you for your question. We thought that audience and researchers can easily trace our experiment using Evian and Contrex because they are available all over the world.

Reviewer 3 Report
In this paper, the authors tried to study the effect of high-temperature ripening in the quality
deterioration of chalky grains through the cell-wall degrading enzymes, such as cellulase
and xylanase.
Authors are recommended to include the equation, in section 2.2, they used to calculate humidity.
Page 3 line 140. Leave a space between the number and the unit (40°C).
Page 3 line 135. Leave a space between the number and the unit (2000g).
Correct the names of the columns in table 4.
Authors are recommended to explain why the standard deviations in Table 4 are too large? It is also recommended that the discussion of these results be expanded.
Author Response
Reviewer3
Comments and Suggestions for Authors
In this paper, the authors tried to study the effect of high-temperature ripening in the quality
deterioration of chalky grains through the cell-wall degrading enzymes, such as cellulase and xylanase.
- Authors are recommended to include the equation, in section 2.2, they used to calculate humidity.
  A: Thank you for your comment. We added equation in L110-L114.
An aluminum cup (Wt) and that of the flour sample (Ws), before heating (Wsb) and af-ter heating (Wsa). Moisture content was calculated as below.
Moisture Content (%) = 100 x ((Wt + Wsb) – (Wt + Wsa))/ Wsb
- Page 3 line 140. Leave a space between the number and the unit (40°C).
A: Thank you for your comment. We added a space between the number and the unit.
- Page 3 line 135. Leave a space between the number and the unit (2000g).
A: Thank you for your comment. We added a space between the number and the unit.
4.Correct the names of the columns in table 4.
A: Thank you for your comment. We corrected the name of columns in table 4.
5.Authors are recommended to explain why the standard deviations in Table 4 are too large?
A: Thank you for your comment. Compared with the other measurements, such as assay of enzyme activities (table 1, 2) or the chemical components (table 5, 6, 7), standard deviations of the measurements of physical properties always show larger standard deviations as shown in our previous report (references No.9, No.12, and No.31).
6.It is also recommended that the discussion of these results be expanded.
A: Thank you for your comment. We revised discussion as below;
As results of the statistical treatments using all the rice samples, xylanase activities of chalky rice grains were shown to be higher than those of whole rice grains (p < 0.01) by Tukey’s one-way analysis of ANOVA, on the contrary, cellulase activities of the chalky grains were shown to be lower than those of whole grains (p < 0.05). (L.300-303)
As results of statistical treatment using all the rice samples, α-amylase and pro-teinase activities of the chalky grains showed significant higher activities than those of whole grains (p < 0.01). Unfortunately, beta-amylase activity did not show the signifi-cant difference between the two rice groups. (L.359-L362)
on the contrary, those of 2 or 4-fold diluted Evian and Contrex did not show the inhi-bition of α-amylase activities. (L384-L387)
but Contrex (0.071 ± 0.0 Ug-1) and 2 or 4- fold dilution of Contrex (0.074 ± 0.0 Ug-1), (0.070 ± 0.0 Ug-1), and those of 2 or 4- fold dilution of Evian (0.064 ± 0.0 Ug-1), (0.066 ± 0.0 Ug-1) did not show any significant effects to endo-xylanase activity. (L.400-403)
Compared with the other measurements, such as assay of enzyme activities (supple-ment Table1, Table 1,), standard deviations of the measurements of physical proper-ties always show larger standard deviations as shown in our previous report (refer-ences No.9, No.12, and No.34). (L439-L442)

Round 2
Reviewer 2 Report
The manuscript has been well revised. But there are still some format errors(such as Reference). Please check the whole manuscript carefully.